# Molecular Basis of Beckwith–Wiedemann Syndrome Spectrum with Associated Tumors and Consequences for Clinical Practice

**DOI:** 10.3390/cancers14133083

**Published:** 2022-06-23

**Authors:** Thomas Eggermann, Eamonn R. Maher, Christian P. Kratz, Dirk Prawitt

**Affiliations:** 1Institute of Human Genetics, Medical Faculty, RWTH Aachen, 52074 Aachen, Germany; 2Department of Medical Genetics, University of Cambridge and Cambridge University Hospitals NHS Foundation Trust, Cambridge CB2 0QQ, UK; erm1000@medschl.cam.ac.uk; 3Paediatric Haematology and Oncology, Hannover Medical School, 30625 Hannover, Germany; kratz.christian@mh-hannover.de; 4Center for Paediatrics and Adolescent Medicine, University Medical Center, 55101 Mainz, Germany; dprawitt@uni-mainz.de

**Keywords:** Beckwith–Wiedemann syndrome spectrum, tumor, genomic imprinting

## Abstract

**Simple Summary:**

Beckwith–Wiedemann syndrome (BWS, OMIM 130650) is an inborn overgrowth disorder caused by molecular alterations in chromosome 11p15.5. These molecular changes affect so-called imprinted genes, i.e., genes which underlie a complex regulation which is linked to the parental origin of the gene copy. Thus, either the maternal gene copy is expressed or the paternal, but this balanced regulation is prone to disturbances. In fact, different types of molecular variants have been identified in BWS, resulting in a variable phenotype; thus, it was consented that the syndromic entity was extended to the Beckwith–Wiedemann spectrum (BWSp). Some molecular subgroups of BWSp are associated with an increased embryonic tumor risk and have different likelihoods for specific tumors. Therefore, the precise determination of the molecular subgroup is needed for precise monitoring and treatment, but the molecular diagnostic procedure has several limitations and challenges which have to be considered.

**Abstract:**

Beckwith–Wiedemann syndrome (BWS, OMIM 130650) is a congenital imprinting condition with a heterogenous clinical presentation of overgrowth and an increased childhood cancer risk (mainly nephroblastoma, hepatoblastoma or neuroblastoma). Due to the varying clinical presentation encompassing classical, clinical BWS without a molecular diagnosis and BWS-related phenotypes with an 11p15.5 molecular anomaly, the syndromic entity was extended to the Beckwith–Wiedemann spectrum (BWSp). The tumor risk of up to 30% depends on the molecular subtype of BWSp with causative genetic or epigenetic alterations in the chromosomal region 11p15.5. The molecular diagnosis of BWSp can be challenging for several reasons, including the range of causative molecular mechanisms which are frequently mosaic. The molecular basis of tumor formation appears to relate to stalled cellular differentiation in certain organs that predisposes persisting embryonic cells to accumulate additional molecular defects, which then results in a range of embryonal tumors. The molecular subtype of BWSp not only influences the overall risk of neoplasia, but also the likelihood of specific embryonal tumors.

## 1. Introduction

Beckwith–Wiedemann syndrome (BWS) is caused by defects in a group of imprinted genes in 11p15.5. Different types of genetic/epigenetic alterations can affect this fine-tuned expression, and result either in an overexpression (e.g., from biallelic expression) or a silencing of imprinted genes. Due to the varying clinical findings of patients with 11p15.5 disturbances, the syndromic entity was extended to the Beckwith–Wiedemann spectrum (BWSp). Several clinical features of BWS overlap with those of other imprinting disorders (i.e., transient neonatal diabetes mellitus (OMIM 601410), Kagami–Ogata syndrome (OMIM 608149), pseudohypoparathyroidism type 1B (OMIM 603233)), and this ambiguity is reflected by molecularly overlapping alterations. Molecular alterations in BWSp patients affect one or both imprinted regions in 11p15.5, which comprise the imprinting control regions 1 and 2 (IC1, IC2). Molecular changes in the same region also occur in patients with Silver–Russell syndrome (OMIM 180860), but in this growth retardation disorder the molecular alterations are opposite to those that occur in BWSp.

## 2. Clinical Features of BWSp

Clinically, the term BWSp describes a complex heterogenous and multisystem disease spectrum. It can be diagnosed by clinical assessment and/or molecular validation. The clinical features comprise characteristic developmental anomalies such as midline abdominal defects (e.g., exomphalos, umbilical hernia), enlarged tongue (macroglossia) and overgrowth (gigantism), lateralized overgrowth (hemihypertrophy) and neonatal hypoglycemia (for review: [1]). 

Up to 8% of all BWSp patients develop an embryonal tumor during their early childhood but the exact tumor risk depends, however, on the precise causative genetic/epigenetic alteration. Thus, depending on the molecular subtype of BWSp, the childhood cancer risk ranges from 1 to 30%. The main tumor types associated with BWSp are Wilms tumor (nephroblastoma), hepatoblastoma and neuroblastoma [1,2]. 

## 3. The Chromosomal Region 11p15.5

The chromosomal region associated with BWSp was initially identified by rare cytogenetic anomalies in BWSp patients involving the distal short arm of chromosome 11 (11p15) [3,4]. The cytogenetic findings involved duplications of 11p13-11p15 [5,6], translocations [7,8,9] and paracentric inversions inv(11) (p11.2p15.5) [10] with the translocation and inversion breakpoints localized in 11p15 [9]. For familial chromosomal anomalies, duplications were found to be of paternal origin and translocations/inversions of maternal origin. In addition, analyses of polymorphic 11p15 markers in BWSp patients suffering from tumors showed preferential maternal-allele loss of 11p15.5 in the tumors [11]. 

Positional cloning efforts led to the identification of several genes in 11p15.5 that were shown to be imprinted and expressed only from the maternally or paternally derived allele. These genes can be grouped into two functionally independent genomic domains (Figure 1). The telomeric domain contains the genes for fetal growth factor *IGF2* (paternally expressed, OMIM 147470) and the long non-coding RNA (lncRNA) *H19* (maternally expressed, OMIM 103280), whereas the centromeric imprinted cluster includes the negative cell cycle regulator gene *CDKN1C* (maternally expressed, OMIM 600856) and the lncRNA *KCNQ1OT1* (paternally expressed, OMIM 604115). The genomic imprinting in these clusters is regulated by parental-specific methylation of two domain-specific differentially methylated regions (DMRs) that mark the respective imprinting control regions (IC1 and IC2).

The IC1 lies within the telomeric imprinting cluster and is controlled by the H19/IGF2:IG DMR between *IGF2* and *H19*. It consists of several repetitively arranged binding sites for the zinc finger protein CTCF (OMIM 604167) and Oct4/Sox2 (OMIM 164177/OMIM 184429). On the paternal allele, methylation of the IC1 differentially methylated region (DMR) prevents CTCF binding and, consequently, leads to expressional silencing of *H19* and expression of *IGF2* on this allele (via specific chromatin looping). The maternal IC1 is unmethylated, thus CTCF binding results in different chromatin looping and *H19* is expressed, whereas *IGF2* is transcriptionally repressed.

The centromeric IC2 is controlled by the KCNQ1OT1:TSS DMR. It is located at the transcriptional start site of the lncRNA *KCNQ1OT1*, an antisense gene of the ion channel KCNQ1 (OMIM 607542). Methylation on the maternal allele inhibits *KCNQ1OT1* expression and enables transcriptional access to *KCNQ1* and *CDKN1C*.

## 4. Molecular Subgroups and Their Causes

In contrast to SRS as its molecular counterpart, BWSp is exclusively linked to molecular alterations in 11p15.5. In BWSp patients, all four types of molecular alterations characteristic for imprinting disorders can be identified (Table 1). These molecular variants comprise genomic alterations (i.e., copy-number variation (CNV), uniparental disomy (upd), single-nucleotide variants (SNV)) and imprinting defects with an impact on the imprinting marks (Figure 1) but without changes of the DNA sequence in the differentially methylated region itself (called imprinting defects or epimutations).

A molecular defect affecting imprinted genes in chromosome region 11p15.5 can be demonstrated in approximately 80% of clinically characterized patients with a clinical diagnosis of BWSp [12]. However, in patients referred for diagnostic genetic testing for BWS, the diagnostic yield is approximately 25% (in preparation).

Imprinting defects (epimutations) in 11p15.5 are the most frequent molecular finding, with a loss of methylation (LOM) at the maternal KCNQ1OT1:TSS DMR allele (found in about 64% of positively tested BWSp cases) (in preparation) associated with transcriptional silencing of *CDKN1C* and presumably *KCNQ1,* as well as transcriptional upregulation of *KCNQ1OT1* [13,14,15].

A subset of BWSp patients with KCNQ1OT1:TSS LOM (12.8%) and single cases with H19/IGF2:IG DMR GOM show aberrant methylation also in other clinically relevant imprinted loci in addition to the two 11p15.5 clusters; these multiple imprinting defects have been called multilocus imprinting defects (MLID) [16].

A gain of methylation (GOM) at the maternal H19/IGF2:IG DMR (IC2) (11.8% of patients) leads to biallelic *IGF2* expression and *H19* silencing [17,18]. In up to 20% of BWSp patients, both imprinting clusters are affected by a mosaic (segmental) paternal uniparental isodisomy (upd(11)pat) of 11p15.5, which occurs postzygotically as a result of mitotic recombination. As both ICs are affected, upd(11)pat results in *IGF2* overexpression and silencing of *H19* as well as *CDKN1C*. Within this cohort, between 4–5% exhibit a mosaic genome-wide paternal UPD, a so-called mosaic paternal uniparental diploidy.

Loss-of-function pathogenic variants in *CDKN1C* (5% of sporadic and 40% of familial cases) and chromosomal copy-number variations in 11p15.5 abolishing the structures and/or protein-binding abilities of either IC can be detected in less than 5% of patients. Due to the suppression of the maternal *KCNQ1OT1* allele, the maternal *CDKN1C* copy is expressed and, therefore, pathogenic CDKN1C variants only have a clinical impact if they are localized on the maternal allele.

Copy-number variants (CNVs) in 11p15.5 account for 2.5% of molecular changes in BWSp patients and the majority comprise duplications of both IC1 and IC2 on the paternal allele. Accordingly, they result in overexpression of paternally expressed genes, whereas expression of maternally expressed imprinted genes in 11p15.5 is unaffected. If the CNV affects additional protein-coding genes, the clinical phenotype might be modified and, thereby, the clinical diagnosis of BWSp might be impeded.

In rare cases, small deletions and duplications have been identified and in these situations the prediction of their functional and clinical impact is difficult, as it is influenced by the size and content of genes and regulative elements. Additionally, the parental origin of the affected allele has to be considered. In this review, the general complexity of 11p15.5 CNVs can only be covered superficially, but for further details specific references are recommended (e.g., [19,20]).

Whereas UPD, CNV and SNV are examples of “classical” genetic disease mechanisms, the molecular cause of imprinting defects (epimutations) is not well understood in the majority of cases. However, environmental factors such as assisted reproduction techniques have been suggested to predispose to aberrant imprinting and the role of genetic causes in some cases is becoming increasingly recognized (for review: [21]). These causes can be divided into genomic alterations affecting (a) the genome of the proband and (b) the maternal genome. Genomic alterations (e.g., CNV or SNV) in the proband can be in cis (i.e., localized either within or close to the aberrant DMR) or in trans (remote from the DMR but with a functional impact on methylation of the DMR). Examples for genomic variants with an in cis effect in 11p15.5 are OCT4/SOX2 binding site alterations affecting the IC1, and *KCNQ1* mutations with an impact on IC2 methylation [22,23]. ZFP57 (OMIM 612192), a protein involved in the maintenance of DNA methylation that is altered in transient neonatal diabetes mellitus, is a notable example of a trans-acting factor associated with multiple epimutations (for review: [24]). However, for BWS, the most frequent genetic causes of MLID are variants in the maternal genome in maternal-effect genes (e.g., *NLRP2* OMIM 609364, *NLRP5* OMIM 609658 and *PADI6* OMIM 610363), which affect components of the oocyte subcortical maternal complex (SCMC). SCMC proteins orchestrate the proper maturation of the oocyte and the early embryogenesis prior to the activation of the fetal genome. Among other functions, some of these proteins are responsible for the maintenance of the imprinting pattern in the early embryo. Accordingly, biallelic maternal variants in these genes are associated with multilocus fetal imprinting defects (for review: [25]).

## 5. (Epi) genotype–Tumor Correlation

BWSp is clinically heterogenous and patients do not necessarily present with all associated features of BWS, nor with a molecular diagnosis [1]. The cardinal feature of a macroglossia can be observed in 90% of all BWSp patients. Midline defects such as exomphalos are also a cardinal feature of BWSp and are preferentially associated with reduced CDKN1C activity due to LOM KCNQ1OT1:TSS DMR or *CDKN1C* pathogenic variants [1]. In some cases, lateralized overgrowth (LO, due to hyperplasia and hypertrophy of affected cells) can be the only detectable phenotypic feature of BWSp (i.e., isolated lateralized overgrowth (ILO, OMIM 235000)) [26]. A diagnosis of BWSp is made in patients who have both ILO and a BWSp-related 11p15.5 molecular defect. Patients with BWSp-ILO are reported to have an increased risk for developing embryonal tumors, mainly nephroblastoma and hepatoblastoma and, despite having a mild BWSp phenotype, should be offered embryonal tumor surveillance according to the molecular subclass for more florid cases of BWSp [27,28,29]. The BWSp molecular subclasses most strongly associated with cancer predisposition are upd(11)pat and GOM at the H19/IGF2:IG DMR [30,31] (Table 2).

The cumulative risk of BWSp patients developing a nephroblastoma is ~5%, but other embryonal tumors including hepatoblastoma, neuroblastoma, adrenal carcinoma and rhabdomyosarcoma occur at lower frequencies. The individual tumor risk depends on the molecular cause of the syndrome. The GOM at the H19/IGF2:IG DMR (IC1) is associated with childhood cancer in 28% of cases (nephroblastoma 24%) and a upd(11)pat has a cumulative childhood tumor risk of approximately 16% (nephroblastoma > hepatoblastoma > neuroblastoma), whereas with a KCNQ1OT1:TSS DMR hypomethylation (IC2 LOM) up to 2.5% of patients develop a tumor (0.7% hepatoblastoma). *CDKN1C* mutations are associated with tumor development in approximately 7% (neuroblastoma > nephroblastoma) [1,32]. Patients with a mosaic paternal uniparental diploidy have a high tumor risk that extends into adulthood. They seem to develop similar types of tumors as patients with upd(11)pat, but with an increased incidence of hepatic and/or adrenal tumors [33]. In the subset of BWSp patients with IC2 LOM and additional epimutations at imprinted DMRs outside of 11p15.5 (i.e., MLID), to date, no specific tumor-risk estimates have been defined [34,35]. The number of detected CNVs causing BWSp to date is too small to enable a reliable conclusion regarding the affected genes/regions and an individual tumor risk.

## 6. General Aspects of the Molecular Basics of Tumors in BWS

BWSp-associated cancers are childhood tumors and are thought to arise during early development. These tumor types are caused by rare mutations that arrest the maturation of cell types (in cases of nephroblastomas in approx. 40 genes) in specific cell populations during restricted developmental windows. In contrast to these tumors, adult cancers arise primarily as a consequence of the accumulation of mutations by transforming mature cells into a cancer cell [36].

Nephroblastomas form the major part of BWSp-associated tumors and arise in the kidney directly from the nephrogenic blastema or from embryonic cells that abnormally persist beyond 36 weeks of gestation (nephrogenic rests) with a preferentially perilobar rather than intralobar histology. During normal development of the kidney, renal vesicles are formed after a mesenchymal to epithelial transformation of the metanephric mesenchyme. These vesicles then expand and differentiate in most of the cell types of the final kidney [37]. Nephroblastomas arise when this process is disrupted, which can occur at different developmental levels, partly reflecting the underlying genetic defects. Depending on the time point of disruption, the resulting tumors contain variable portions of blastemal, epithelial and stromal cells. Nephrogenic rests may terminate their differentiation, regress (e.g., become sclerotic), or form hyperplastic, oncogenic cells.

Nephrogenic rests seem to carry fewer mutations than their adjacent nephroblastoma [38], supporting the idea that they are only predisposed cells that need the acquisition of further molecular alterations to result in a tumor. BWSp patients with nephroblastoma that show loss of imprinting (LOI) at 11p15, affecting *IGF2* and *H19* either by GOM at the IC1 or upd(11)pat, have a high frequency (17.3%) of bilateral tumors [39], suggesting that the overexpression of *IGF2* is causative for the nephrogenic rests. Consequently, the BWSp tumors have additional mutations, e.g., in the transcription factor *WT1* (OMIM 194070), in the Wnt pathway regulators beta-catenin (*CTNNB1*, OMIM 116806) and AMER1 (*WTX*, OMIM 300647) or in the apoptosis master regulator *TP53* (OMIM 191170). The mutational spectrum also includes genes involved in miRNA biogenesis (*DROSHA* OMIM 608828, *DICER1* OMIM 606241, *DIS3L2* OMIM 614184, *SIX1/2* OMIM 601205/OMIM 604994 or *MYCN* OMIM 164840) or in chromatin silencing (*TRIM28,* OMIM 601742). The specific combination of adverse effects of LOI and gene mutation is partly reflected in the histological subtype of the resulting nephroblastoma (Figure 1). *TRIM28* mutations are often observed in epithelial predominant tumors [40], whereas *TP53* pathogenic variants are strongly associated with anaplastic nephroblastoma and *CTNNB1/WT1* pathogenic variants with the stromal phenotype. A mixed, blastemal tumor is more likely to have a pathogenic variant in a gene of the miRNA biogenesis (reviewed in [41]).

The second most common tumor entity associated with BWSp is hepatoblastoma, which is mainly observed in upd(11)pat cases. In addition to the suggestive *IGF2* overexpression, which might also lead to cessation of cellular maturation in the developing liver, additional genetic alterations seem to drive tumor formation. In patients with non-BWSp-associated hepatoblastoma, SNPs in the *H19* gene have been associated with decreasing or increasing hepatoblastoma risk (reviewed in [42]). The loss of *H19* expression in the mosaic upd(11)pat situation and/or an unfavorable SNP combination might then promote tumor development in IGF2 overexpressing persisting embryonic liver cells.

Neuroblastomas are the third major BWSp-associated tumor type, occurring in BWSp with IC1 GOM and upd(11)pat, but also in the subgroup with *CDKN1C* loss-of-function mutations. Neuroblastomas are neuroendocrine tumors that originate from the developing sympathetic nervous system, resulting in tumor development localized most frequently in the adrenal glands and sympathetic ganglia. The most common causative genomic alteration observed here is a *MYCN* gene amplification, which is present in approximately 18% of all cases, that can transcriptionally upregulate the histone-lysine N-methyltransferase enzyme *Enhancer of zeste homolog 2* (*EZH2*, OMIM 601573), promoting an undifferentiated neuroblastoma tumor phenotype associated with poor clinical outcomes ([43], reviewed in [44]). When *MYCN* was knocked-down by RNAi in selected neuroblastoma cell lines, cyclin A1 (*CCNA1*, OMIM 604036) and cyclin G2 (*CCNG2*, OMIM 603203) were upregulated alongside the cyclin-dependent kinase inhibitor *CDKN1C* [45], suggesting a functional connection between MYCN amplification and cell cycle control. High *MYCN* expression results in low *CDKN1C* expression. BWSp neuroblastomas either have an overexpression of *IGF2* (IC1 GOM or upd(11)pat) and/or *CDKN1C* reduction (upd(11)pat or *CDKN1C* loss-of-function mutations), which might be the relevant “first hit” for parts of the sympathetic nervous system to pave the road for neuroblastoma development due to a second alteration such as *MYCN* amplification.

The epigenetic alterations in 11p15.5 (LOI, LOH) associated with BWSp can also be found in a number of tumors in patients without BWS. In non-syndromic nephroblastoma, constitutional 11p15.5 CNV resulting in H19/IGF2:IG DMR hypermethylation could be detected [46]. In these tumors, the same abnormality also identified in lymphocytes was present and the level of H19/IGF2:IG DMR methylation was greater in the tumor, suggesting a causative role of the GOM in driving oncogenesis. This consideration is also supported by a study that analyzed whether a Wilms tumor may evolve from precancerous clonal expansions in the kidney (47). They found premalignant somatic mutations (clonal expansions), shared between tumor and normal tissue but absent from blood, in morphologically normal kidney tissues that preceded tumors. Of these clonal expansions, 58% were GOM of H19/IGF2:IG DMR. The group then compared somatic changes of nephroblastoma with and without clonal nephrogenesis. In tumors with clonal nephrogenesis, LOH of 11p15 was mostly absent, indicating that there may be two distinct pathways for nephroblastoma generation. Both pathways utilize dysregulation of 11p15 genes as a driver of oncogenesis, either resulting in a tumor arising directly in isolation through LOH of 11p15, or indirectly via clonal nephrogenesis with perturbation of 11p15 by GOM of H19/IGF2:IG DMR [47].

A LOI at 11p15.5 (IGF2:TSS LOM) was detected in breast (33%) and colorectal (80%) tumor tissues [48]. Since these tumors are adult tumor types, it can be speculated that the LOI of 11p15.5 in these cases represents a secondary somatic alteration, occurring in cells that already were dedifferentiated in the tumor progression due to an inherited or an acquired tumor-type-specific genetic mutation. Pyrosequencing analysis of DMRs in corresponding prostate and prostate cancer tissue samples revealed frequent hypo- and hypermethylation at the H19/IGF2:IG DMR in both benign and cancerous tissues [49]. Consequent expression analyses in the prostate cancer tissues showed a significantly diminished expression of the imprinted genes *PLAGL1/ZAC1* (OMIM 603044), *MEG3* (OMIM 605636), *NDN* (OMIM 602117), *CDKN1C, IGF2*, and *H19*, whereas *KCNQ1OT1* was significantly overexpressed [49]. The expression level of many of these genes was strongly correlated, suggesting the existence of an imprinted gene network (IGN) previously reported in mice [50]. The deregulation of the IGN also correlated with EZH2 and HOXC6 (OMIM 142972) overexpression, similar to the findings in neuroblastomas described above. It is tempting to speculate that certain disturbances due to causal molecular findings in BWSp subgroups maintain embryonic states of cells and stall further cellular differentiations due to altered transcriptional dosage of co-expressed genes in the IGN. The lncRNA *H19* has been shown to influence the transcription of at least five other genes of the IGN (including *Igf2* and *Peg1* OMIM 601029) by recruiting the methyl-CpG-binding domain protein 1 (MBD1, OMIM 156535) that transcriptionally downregulates target genes by bringing repressive histone marks to the differentially methylated regions of these targets [51]. This could also explain why different BWSp alterations result in similar tumor predispositions. The presumable main effectors would then be loss-of-function/expressional reduction of *CDKN1C, KCNQ1OT1* overexpression, *IGF2* overexpression and *H19* reduction. In mammalians CDKN1C has a conserved, likely antagonistic function to IGF2 in the placenta [52]. This further supports the idea that transcriptional alterations in either IC could have comparable effects on developmental pathways. In adult tumor types, an IGN disturbance affecting these genes in a suitable way could be either a first or a second hit for tumor formation.

## 7. Relevance for Molecular Diagnostic Testing

The molecular heterogeneity of BWSp requires a comprehensive molecular testing approach. By considering the frequencies of the molecular subtypes of 11p15.5 variants, methylation tests addressing specific CpGs in the H19/IGF2:IG DMR and the KCNQ1OT1:TSS DMR have been recommended as first step analyses [1]. These tests allow the detection of imprinting defects, CNVs and upd(11)pat in the same analysis, and depending on the method, further assays to discriminate between the different molecular subtypes might be applied. In case of negative test results, *CDKN1C* sequencing can be conducted.

However, the detection of imprinting defects and upd(11)pat can be hampered by mosaicism; in particular, upd(11)pat might escape detection because of low-level mosaicism (e.g., [29]). Thus, in case of a strong clinical suspicion of BWSp but a negative testing result from peripheral blood, analysis of other tissues should be considered, preferably a biopsy from an organ which is affected by the disease (e.g., from hyperplastic organ or tumor).

## 8. Relevance for Clinical Management

The initial treatment, histological classifications and staging criteria of nephroblastoma differs, depending on the geographical location of the patient. Two major treatment strategies have been established, one from the International Society of Pediatric Oncology (SIOP) and the second from the Children’s Oncology Group (COG).

The SIOP classification of nephroblastomas consists of three major risk groups ranging from low-risk tumors (completely necrotic), intermediate-risk tumors (stromal, epithelial) and high-risk tumors (blastemal or diffuse anaplastic). According to the SIOP protocol, infants younger than 6 months of age that suffer from a unilateral nephroblastoma obtain an initial—preferentially partial—nephrectomy. All patients with a suspected nephroblastoma older than 6 months of age receive either 4 weeks of preoperative chemotherapy with actinomycin D and vincristine (if the tumor is localized) or 6 weeks of actinomycin D, vincristine and doxorubicin (if the tumor is already metastatic). A percutaneous core needle biopsy is suggested in children 7 years of age and older or children with an uncertain clinical picture to avoid a misdiagnosis of nephroblastoma. After the neoadjuvant chemotherapy, a radical nephrectomy is performed in most BWSp patients with unilateral WT if the preoperative chemotherapy does not support a nephron-sparing surgery (NSS), which is also applied in case of bilateral nephroblastomatosis or WTs [41].

The COG classification of WTs distinguishes anaplastic (focal and diffuse) and non-anaplastic (favorable histology) WTs. The COG strategy favors a primary (partial) nephrectomy for unilateral renal masses in patients without WT predisposition, but in BWSp a neoadjuvant chemotherapy (without biopsy) with the aim of preserving renal units and subsequent surgery at 6–12 weeks after initiation of chemotherapy is suggested [53].

After surgery, according to SIOP and COG, a risk-based adjuvant chemotherapy and radiotherapy is administered with the help of multiple prognostic factors [54].

Hepatoblastoma (HB) arise from a hepatocyte precursor cell and consists of either epithelial (epithelial HB) or epithelial and mesenchymal elements (mixed HB). In addition, 80–90% of HB are observed in children between 6 months and 5 years with a median age of 18 months [55]. Treatment of syndrome-associated HB is not different to that of sporadic HB. Epithelial HB presents with histologically well-differentiated fetal cells and low mitotic activity. It defines the group of very-low-risk patients that are treated with upfront surgery only, without the need for further therapy if the tumor is completely resected [56]. If the tumor is unresectable, or if a mixed HB is diagnosed, a neoadjuvant chemotherapy with cisplatin components precedes surgery, followed by an adjuvant chemotherapy with mixed cytostatic drugs to avoid a resistance against cisplatin. The COG, the International Childhood Liver Tumors Strategy Group (SIOPEL), the Society for Pediatric Oncology and Hematology (GPOH), and the Japanese Pediatric Liver Tumors Group (JPLT) use different platinum-based chemotherapy regimens with similar success [57]. In case of unresectable HB or if the HB is insensitive to the chemotherapeutics, a liver transplantation can be considered. Radiation therapy has hardly any effect on HB and is therefore not considered, but, for example, a transarterial radioembolization with yttrium-90 as adjunctive therapy may be used to bridge to surgical resection or liver transplantation [57].

Neuroblastoma most likely originates from sympathoadrenal progenitor cells within the neural crest that normally differentiate to sympathetic ganglion cells and adrenal catecholamine-secreting cells [58]. MYCN amplification can affect all tumor cells (homogenous MYCN amplification, homMNA), which consequently is associated with a poor prognosis. Alternatively, it can affect only part of the tumor cells (hetMNA). Interestingly, upd(11)pat is almost exclusively detected in tumors of younger patients and only very rarely observed in homMNA tumors [59]. This makes an argument for the concept that BWSp causative alterations stall the differentiation of sympathoadrenal progenitor cells and eventually result in hetMNA neuroblastomas due to clonal second hits.

Currently, neuroblastoma occurring in BWSp patients are treated like isolated neuroblastoma. Treatment depends on a stratification system that aims to group the patients in different risk groups according to tumor-free survival after treatment. To establish a consensus approach for pretreatment risk stratification, an International Neuroblastoma Risk Group (INRG) task force representing the major pediatric cooperative groups around the world was formed in 2005 [60]. The risk classification (very low risk, low risk, intermediate risk, or high risk) is based primarily on age of the patient (younger or older than 18 months), International Neuroblastoma Staging System (INSS) stage [61], MYCN status, ploidy of tumor cells, and Shimada histology. Low-risk neuroblastomas have a tendency to spontaneously regress, eventually supported by application of 13-cis-retinoic acid, which induces differentiation and which is currently also used as maintenance therapy to avoid residual disease in patients with high-risk disease. Patients with high-risk neuroblastoma (nearly half of cases of neuroblastoma) are typically treated with four therapeutic phases: first, a neoadjuvant induction therapy, second, surgical removal of the tumor followed by radiation therapy, third, a consolidation phase with single or tandem high-dose chemotherapy (with or without metabolic irradiation by 131I-MIBG or targeted treatments) and autologous transplant and, finally, fourth is immunotherapy to eliminate residual disease [62,63].

## 9. Relevance for Follow-Up

To date, no large series of adults with BWSp have been examined for health issues in later life. The consensus group agreed that BWSp molecular subgroups and patients with classical BWS and no detectable molecular anomaly should be offered an abdominal ultrasound (USS) every 3 months until the age of 7 years, and only BWSp patients with IC2 LOM should not be offered routine USS [1]. This covers the relevant phase of life regarding the BWSp-associated embryonal neoplasias. With the exception of BWSp patients with mosaic paternal uniparental diploidy, no obvious association of BWSp and predisposition to adult-onset carcinomas/rare endocrine tumors has been reported; thus, there is no evidence of a specific tumor risk that might justify surveillance over the age of 7 years. However, similar to children with sporadic tumors, children with BWSp who are treated for embryonal tumors might develop late-onset complications from therapy.

## 10. Conclusions

The clinical heterogeneity of BWSp and the wide underlying spectrum of molecular alterations present a challenge for clinical and molecular diagnosis. Nevertheless, the precise determination of the molecular subgroup is required as the basis for personalized monitoring and treatment, especially in regard to individual tumor risk. Tumors occurring in BWSp patients do not exactly resemble the corresponding tumor types in non-syndromic patients. They have different combinations of causative (epi)genetic alterations, sometimes reflected in the histology of the resulting neoplasia. Thus, an improved molecular diagnosis of BWSp might lead to further optimized (tumor-)treatment protocols based on the understanding of personalized tumor-driving pathways.

## Figures and Tables

**Figure 1 cancers-14-03083-f001:**
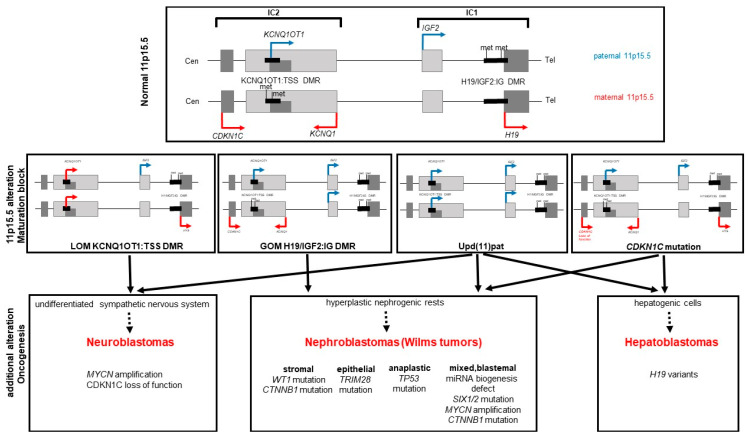
Schematic representation of genetic and epigenetic defects causing BWSp and tumor types associated with the specific defects. *Upper Panel:* Schematic delineation of the chromosomal region 11p15.5 with the two imprinting clusters IC1 and IC2, the allele-specific methylation of the respective DMRs and the consequent expression of imprinted genes on the maternal (red) and paternal (blue) allele. *Middle Panel:* Defects observed in BWSp patients with resulting transcriptional consequences, leading to a maturation block in cells of susceptible organs. These cells may persist after birth and then either terminate their differentiation, regress or progress to form hyperplastic and tumorigenic cells. *Lower Panel:* Tumor types and histological subclasses that occur with certain risk due to typical secondary mutations in the cells that were stalled in development.

**Table 1 cancers-14-03083-t001:** Frequencies of the different molecular subtypes among molecularly confirmed patients with BWSp based on a recent survey from eleven diagnostic labs from seven countries (in preparation). Frequency of the mutation detection rates are not given as *CDKN1C* sequencing is not conducted routinely in all patients with the clinical diagnosis of BWS (For abbreviations: see text).

	Molecular Subtype	Total Number (n = 1301)	Ratio among Solved Cases
expected molecular diagnoses	IC1 GOM	153	11.8%
	IC2 LOM	833	64.0%
	*of these MLID:* *12.8%*	*107*	
	CNVs 11p	32	2.5%
	upd(11)pat	254	19.5%
	*of these uniparental* *diploid: 4.3%*	*11*	
unexpected molecular diagnoses	IC1 LOM	27	2.1%
	PHP	2	0.2%

**Table 2 cancers-14-03083-t002:** Percentage of cancer risk for each molecular subgroup (according to [1]).

Molecular Defect	Frequency of Molecular Defect	Tumor Risk (% of Patients)
**IC1 GOM**	5%	*Overall risk (28.1%)*
		Wilms tumor (24%)
		Neuroblastoma (0.7%)
		Pancreatoblastoma (0.7%)
**IC2 LOM**	50%	*Overall risk (2.6%)*
		Hepatoblastoma (0.7%)
		Rhabdomyosarcoma (0.5%)
		Neuroblastoma (0.5%)
		Thyroid cancer (0.3%)
		Wilms tumor (0.2%)
		Melanoma (0.1%)
**upd(11)pat**	20% (see also paternal uniploidy)	*Overall risk (16%)*
		Wilms tumor (7.9%)
		Hepatoblastoma (3.5%)
		Neuroblastoma (1.4%)
		Adrenocortical carcinoma (1.1%)
		Pheochromocytoma (0.8%)
		Lymphoblastic leukemia (0.5%)
		Pancreatoblastoma (0.3%)
		Hemangiotheloma (0.3%)
		Rhabdomyosarcoma (0.3%)
**Loss-of-function CDKN1C variants**	5% (40% in familial cases)	*Overall risk (6.9%)*
		Wilms tumor (1.4%)
		Neuroblastoma (4.2%)
		Acute lymphatic leukemia (1.4%)

Italics to distinguish from individual tumor-types; bold for better recognition.

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
