# Peer review of "Molecular Basis of Beckwith–Wiedemann Syndrome Spectrum with Associated Tumors and Consequences for Clinical Practice"

_cancers, 2022, doi:10.3390/cancers14133083_

Round 1
Reviewer 1 Report
Adequate review but I must make some comments.
OMIM numbering of the genes described, in the summary the OMIM numbering for the entity under study can also be used.
Numerous keywords, I think it can be simplified.
The main comment I have is that as a review article its content is a bit dense and different tables can improve its presentation. That would improve your understanding.
There are current references not used in this article, some genes in the references do not appear in italics and the standards should be reviewed, some references only appear by year, for example.
Author Response
Suggestions for Authors
Adequate review but I must make some comments.
OMIM numbering of the genes described, in the summary the OMIM numbering for the entity under study can also be used.
ANSWER: Done
Numerous keywords, I think it can be simplified.
ANSWER: Done
The main comment I have is that as a review article its content is a bit dense and different tables can improve its presentation. That would improve your understanding.
ANSWER: A further table 2 about tumor risks has been added.
There are current references not used in this article, some genes in the references do not appear in italics and the standards should be reviewed, some references only appear by year, for example.
ANSWER: We have reviewed the paper for these aspects.

Reviewer 2 Report
In my opinion, this is a well-written and interesting manuscript that describes the most frequent neoplasia associated with the Beckwith-Wiedemann syndrome spectrum (BWSp) phenotype and the suspected molecular causes of abnormal growth, mostly linked to gene dosage alterations of imprinted genes. The review allows an easy understanding of the (epi)genetic bases of this heterogeneous disorder and provides an update on tumor risk within each molecular subgroup.
Few minor observations:
· Can the authors summarize in a table the percentage of cancer risk for each molecular subgroup? This can help to elucidate unclear points. For instance, if the total risk for the IC2LOM group was reported to be 2.5%, of which 0.7% hepatoblastoma, is 1.8% the risk of neuroblastoma, the most frequent neoplasia in this group?
· Is the reference 16 the right one?
· Few typos can be found, e.g. in Abstract, first line: BWS not “BWSp”; in paragraph 3, first line, double “rare”.
Author Response
In my opinion, this is a well-written and interesting manuscript that describes the most frequent neoplasia associated with the Beckwith-Wiedemann syndrome spectrum (BWSp) phenotype and the suspected molecular causes of abnormal growth, mostly linked to gene dosage alterations of imprinted genes. The review allows an easy understanding of the (epi)genetic bases of this heterogeneous disorder and provides an update on tumor risk within each molecular subgroup.
Few minor observations:
- Can the authors summarize in a table the percentage of cancer risk for each molecular subgroup? This can help to elucidate unclear points. For instance, if the total risk for the IC2LOM group was reported to be 2.5%, of which 0.7% hepatoblastoma, is 1.8% the risk of neuroblastoma, the most frequent neoplasia in this group?
ANSWER: A further table 2 about tumor risks has been added.
- Is the reference 16 the right one?
ANSWER: The reviewer is right, the reference has been removed.
- Few typos can be found, e.g. in Abstract, first line: BWS not “BWSp”; in paragraph 3, first line, double “rare”.
ANSWER: Checked.

Reviewer 3 Report
Thank you for the opportunity to review the manuscript “
Molecular basis of Beckwith-Wiedemann syndrome spectrum with associated tumors and consequences for clinical practice” by Thomas Eggermann et al. Beckwith Wiedemann Syndrome (BWS) is a rare, genetically and clinically
heterogeneous disorder. The authors report on tumor risk in patients with different BWS genotypes .
I find the paper interesting and valuable for both geneticists and clinicians.
However, there are some points that need to be improved.
1. Firstly, there is a need for a clarification of BWS and BWSp. It should be done in the Intro-duction section rather than later in the text. Still the OMIM entry is for BWS (# 130650). The Authors should be consistent and the use of BWS and BWSp should be clear through-out the text, as it is sometimes mixed, e.g. “2. Clinical features of BWS. Clinically, the term BWSp describes a complex heterogenous and multisystem disease spectrum.”
2. Please shorten Simple Summary as the information is overlapped with Abstract.
3. 5.(. Epi)genotype phenotype correlation it is ra ther a description of a risk for tumor de-
velopment, no t the whole phenotype ““(. Epi)genotype m isspelling
4. 6. General aspects of the molecular basics of tumors in BWS this section is too long it
sh ould mainly des cribe the details regarding BWS genotypes
5. 8. Relevance for clinical management too long , not clearl y informative for specific man-
agement /recomme nded follo w up for patients with BWS.
6. I woul d recommend presenting a Table with d ifferent genotype s and possible phenotype/ risk
for tumor development
Author Response
Thank you for the opportunity to review the manuscript “Molecular basis of Beckwith-Wiedemann syndrome spectrum with associated tumors and consequences for clinical practice” by Thomas Eggermann et al. Beckwith Wiedemann Syndrome (BWS) is a rare, genetically and clinically heterogeneous disorder. The authors report on tumor risk in patients with different BWS genotypes .I find the paper interesting and valuable for both geneticists and clinicians. However, there are some points that need to be improved.
- Firstly, there is a need for a clarification of BWS and BWSp. It should be done in the Intro-duction section rather than later in the text. Still the OMIM entry is for BWS (# 130650). The Authors should be consistent and the use of BWS and BWSp should be clear through-out the text, as it is sometimes mixed, e.g. “2. Clinical features of BWS. Clinically, the term BWSp describes a complex heterogenous and multisystem disease spectrum.”
ANSWER: Done
- Please shorten Simple Summary as the information is overlapped with Abstract.
ANSWER: Done
- 5.(. Epi)genotype phenotype correlation it is ra ther a description of a risk for tumor de-
velopment, no t the whole phenotype ““(. Epi)genotype m isspelling
ANSWER: Done

Round 2
Reviewer 3 Report
Please see the attached review.

Author Response
Reviewer 3:
Thank you for improving the manuscript. Please check the BWS and BWSp use, to make it accurate.
Please correct e.g.:
Abstract: line 37- “Beckwith-Wiedemann syndrome” not “Beckwith-Wiedemann syndrome spectrum”.
Answer: Done
The description of BWS vs BWSp should be included as well in the Introduction (as in Summary).
Answer: Done
Line 68- “Clinical features of BWSp” not “Clinical features of BWS”
Answer: Done
